# Advancements in Superhydrophobic Paper-Based Materials: A Comprehensive Review of Modification Methods and Applications

**DOI:** 10.3390/nano15020107

**Published:** 2025-01-12

**Authors:** Yin Tang, Shouwei Ban, Zhihan Xu, Jing Sun, Zhenxin Ning

**Affiliations:** 1College of Light Industry and Textile, Qiqihar University, Qiqihar 161006, China; 03245@qqhru.edu.cn (Y.T.); xuxpao@163.com (Z.X.); 2College of Chemistry and Chemical Engineering, Qiqihar University, Qiqihar 161006, China; 2024913333@qqhru.edu.cn; 3Engineering Research Center for Hemp and Product in Cold Region of Ministry of Education, Qiqihar 161006, China; 4Department of Academic Theory Research, Qiqihar University, Qiqihar 161006, China; 03292@qqhru.edu.cn; 5Qiqihar Inspection and Testing Center, Qiqihar 161006, China

**Keywords:** superhydrophobic, paper-based functional material, oil–water separation, self-cleaning

## Abstract

Superhydrophobic paper-based functional materials have emerged as a sustainable solution with a wide range of applications due to their unique water-repelling properties. Inspired by natural examples like the lotus leaf, these materials combine low surface energy with micro/nanostructures to create air pockets that maintain a high contact angle. This review provides an in-depth analysis of recent advancements in the development of superhydrophobic paper-based materials, focusing on methodologies for modification, underlying mechanisms, and performance in various applications. The paper-based materials, leveraging their porous structure and flexibility, are modified to achieve superhydrophobicity, which broadens their application in oil–water separation, anti-corrosion, and self-cleaning. The review describes the use of these superhydrophobic paper-based materials in diagnostics, environmental management, energy generation, food testing, and smart packaging. It also discusses various superhydrophobic modification techniques, including surface chemical modification, coating technology, physical composite technology, laser etching, and other innovative methods. The applications and development prospects of these materials are explored, emphasizing their potential in self-cleaning materials, oil–water separation, droplet manipulation, and paper-based sensors for wearable electronics and environmental monitoring.

## 1. Introduction

Surfaces exhibiting superhydrophobicity are characterized by their remarkable ability to repel water, with water contact angles (WCA) surpassing 150° and sliding angles (SA) remaining under 5° [1]. These surfaces draw inspiration from natural models like the lotus leaf, which demonstrates self-cleaning characteristics owing to its micro–nano hierarchical architecture and reduced surface energy [2]. The superhydrophobic state can be achieved by combining low-surface-energy materials with micro/nanostructures that create air pockets, preventing water from spreading and maintaining a high contact angle [3,4]. The definition is crucial for understanding the unique properties and potential applications of such surfaces in various fields.

Paper-based functional materials have gained significant attention due to their renewable, biodegradable, and sustainable nature. They offer a wide range of applications in areas such as packaging [5], medical diagnostics [6], and environmental protection [7]. The versatility of paper stems from its porous structure, which can be easily modified to achieve specific functionalities, such as superhydrophobicity [8]. This property enhancement broadens the application scope of paper-based materials, making them suitable for oil–water separation, anti-corrosion, and self-cleaning applications [9,10,11].

In today’s society, environmental pollution is becoming increasingly severe, with oil–water pollution posing a significant threat to the ecological environment and human health. Traditional oil–water separation technologies suffer from low efficiency, high costs, and secondary pollution, making it difficult to meet the increasingly stringent environmental protection requirements. Consequently, there is an urgent need for the development of new and efficient oil–water separation materials. Superhydrophobic paper-based functional materials, with their unique surface structures and excellent hydrophobic properties, offer a promising solution for efficient oil–water separation. These materials provide new insights and methodologies for addressing environmental challenges, highlighting their practical significance and broad application potential. Moreover, as technology continues to advance, the demand for multifunctional materials is growing. Superhydrophobic paper-based materials have emerged as a viable option in various fields, including smart packaging and environmental monitoring, offering substantial support for the evolution of related industries [12]. For example, the application of materials with special conductivity and surface chemical properties such as Mxene in superhydrophobic paper-based materials makes the application of this material [13,14,15].

In the realm of superhydrophobic paper-based materials, innovation in preparation methods is crucial for enhancing performance and expanding functionality. Recent advancements have seen the integration of diverse cutting-edge technologies, such as laser etching and physical compositing. Laser etching technology, for instance, enables the precise creation of micro- and nanostructures on paper surfaces, thereby accurately controlling surface wettability [16]. This advancement paves the way for the development of high-performance paper-based sensors and microfluidic devices. Additionally, the convergence of superhydrophobic paper-based materials with sensor technologies has led to the creation of novel paper-based strain sensors. These sensors are capable of monitoring human movement and detecting underwater vibrations, representing a pioneering application in the fields of wearable electronics and underwater robotics. Such multifunctional applications are at the forefront of innovation, expanding the horizons of superhydrophobic paper-based materials and underscoring the dynamic progress in this research area [17].

The primary objective of this review is to provide a comprehensive overview of recent advancements in the development of superhydrophobic paper-based functional materials and systematically classify the latest preparation techniques in recent years, encompassing an in-depth analysis of the methodologies employed for superhydrophobic modification, elucidation of the underlying mechanisms driving these modifications, and evaluation of the resulting material’s performance across diverse applications.

## 2. Paper-Based Functional Materials

Paper-based functional materials are a class of materials that harness the inherent properties of paper, such as its porous structure and flexibility, to serve specific functional purposes beyond traditional uses [9,18,19]. These materials are defined by their ability to interact with their environment in a controlled manner, which is achieved through chemical modifications or the introduction of functional components [19]. The properties of paper-based functional materials can be tailored to achieve desired characteristics such as superhydrophobicity, enhanced strength, or specific chemical reactivity [20].

Paper is primarily composed of cellulose fibers, which give it its unique set of physical and chemical properties. The porous nature of paper allows for the absorption and transportation of liquids, making it an ideal substrate for functionalization [21]. The hydrophilic character of cellulose can be altered through chemical treatments, enabling the development of superhydrophobic surfaces [22]. Additionally, paper’s mechanical properties, such as its strength and flexibility, can be enhanced through cross-linking agents or by incorporating nanomaterials [23]. The surface properties of paper can be tailored by introducing nanostructures or by applying chemical coatings [24]. Nanoscale roughness can be created through various methods, including the deposition of nanoparticles or the use of nanolithography techniques [25]. These structures increase the surface area and promote air pocket formation when exposed to water, contributing to superhydrophobic behavior [26].

## 3. Applications of Paper-Based Functional Materials

In the realm of material science, the advent of paper-based functional materials has marked a significant shift towards sustainable and versatile solutions across various industries. Traditionally known for its use in writing, printing, and packaging, the paper has now been repurposed and engineered to serve in cutting-edge applications that address contemporary challenges [18,19]. From healthcare diagnostics to environmental conservation, and from energy generation to food safety, the utility of paper-based materials is expanding at an unprecedented rate [27,28,29,30,31,32,33,34,35,36,37]. The inherent properties of paper, such as its porous structure, flexibility, and biodegradability, make it an ideal candidate for functionalization [21]. Scientists and engineers have capitalized on these attributes to create materials that can perform complex tasks while maintaining an eco-friendly profile. The ability to transform a readily available and renewable resource into a platform for high-tech applications is not only innovative but also aligns with the global push toward sustainability.

Diagnostics

Paper-based diagnostic platforms have emerged as a game-changer in global public health, particularly for point-of-care testing in low-resource settings [27]. Martinez et al. [6] developed the use of patterned paper for low-volume, portable bioassays, demonstrating the potential for paper to serve as an inexpensive substrate for diagnostic tests (Figure 1). The simplicity and disposability of paper make it an attractive medium for lateral flow assays, which are widely used for the rapid detection of various analytes, including infectious diseases and chemical contaminants [28]. Advances in paper-based diagnostics have been marked by the development of microfluidic devices that can perform complex analyses with minimal sample volumes [29,30]. These devices provide significant benefits, including facile miniaturization and integration, low sample and reagent requirements, and rapid reaction kinetics, offering a novel approach to point-of-care testing (POCT) [30].

Environmental Management

In environmental management, paper-based materials have been engineered for the treatment of complex wastewater, which contains both oil and heavy metal ions. These materials, derived from cellulose, are prepared through a green papermaking process and exhibit high wet strength and excellent oil–water separation properties. The introduction of amphoteric functional groups, such as amino and carboxyl, enhances their performance across a broad pH range, making them suitable for continuous and efficient purification of complex wastewater (Figure 2) [31].

Energy Generation

Paper-based materials have also been harnessed for energy generation, particularly in the development of biofuel cells. Gao et al. [32] described the use of untreated print paper to generate electricity under moisture ingress, showcasing the potential of paper-based materials in moist–electric power generation. An untreated print paper sample (with an area of 1.5 cm^2^) can generate a voltage of 0.25 V and a current of 15 nA. The power output can be easily adjusted by modifying environmental factors such as humidity and temperature, as well as by varying the number of devices through simple series or parallel connections. These paper-based moist–electric generators (PMEGs) are anticipated to have practical applications in everyday ambient environments due to the widespread availability and cost-effectiveness of paper materials.

Food Testing

The application of paper-based sensors in the realm of food testing is a burgeoning domain, offering cost-effective and user-friendly solutions for the detection of food contaminants [33]. Furthermore, the development of paper-based sensors is not confined to detecting a single type of pollutant.

In practical applications, the portability and ease of use of paper-based sensors make them ideal for on-site rapid testing. They can be designed as disposable test strips or portable detection kits, allowing food producers, regulators, and consumers to conduct quick tests anywhere. However, despite the great potential of paper-based sensors in food testing, there are still challenges to overcome. Enhancing the sensitivity and specificity of sensors to ensure accurate detection of low-concentration contaminants is a crucial research direction. Additionally, developing sensors that can operate stably in complex food matrices, as well as improving production efficiency and reducing costs, are also key focuses of current research. With continuous technological advancements, paper-based sensors are expected to play an increasingly important role in food safety and quality control. These sensors can be customized to identify specific pollutants such as pesticide residues and microbial contamination. For instance, researchers have developed paper-based electrochemical biosensors for detecting ethanol levels in beer [34], as well as inkjet-printed flexible biosensors for the rapid, label-free detection of antibiotics in milk [35]. These applications underscore the practical value of paper-based sensors in ensuring food safety and quality control.

Smart Packaging

Paper-based functional materials have emerged as a key component in smart packaging due to their sustainability and versatility. These materials are used for various applications, including antimicrobial, antioxidant, and moisture control systems (Figure 3) [36]. They are also integrated into ethylene removal systems to extend the shelf life of fresh produce, as ethylene is a ripening hormone that can degrade the quality of fruits and vegetables. Smart packaging systems with embedded sensor technology are employed to monitor food freshness and extend shelf life, improving product safety and quality standards. The use of natural biopolymers in packaging materials, such as protein, starches, and cellulose, with biological activities like antibacterial and antioxidants, regulates the environment inside the packaging. Intelligent packaging materials can monitor the condition of packaged food or the environment surrounding the food, communicating the product’s history to consumers without interacting with the food [37]. These advancements in paper-based functional materials for smart packaging showcase their potential to enhance food preservation and safety, aligning with the growing demand for sustainable and intelligent packaging solutions.

## 4. Superhydrophobic Modification of Paper-Based Materials

The pursuit of superhydrophobic properties on paper-based materials has led to significant advancements in the field of materials science. Superhydrophobicity, characterized by an extreme aversion to water, is a natural phenomenon exemplified by the lotus leaf [38]. This trait is captivating and possesses a wide range of applications in industries such as manufacturing, construction, and textiles. The techniques for achieving superhydrophobicity on paper involve chemical modifications that reduce surface energy and enhance the water contact angle, physical modifications that create micro- and nanostructures to increase surface roughness, and hybrid approaches that combine both methods for enhanced durability and robustness [39]. These strategies are inspired by natural structures and utilize a variety of materials, including low-surface-energy chemicals, nanoparticles, and polymers. The integration of these techniques has resulted in surfaces that mimic the water-repelling effect seen in nature, offering innovative solutions for various practical applications.

### 4.1. Superhydrophobic Materials

Superhydrophobic materials, known for their exceptional water-repelling abilities, have been the subject of intense research due to their potential applications in anti-corrosion, self-cleaning surfaces, and oil–water separation [40]. These materials are designed to mimic natural surfaces such as the lotus leaf (Figure 4), which exhibits micro/nano hierarchical structures that promote water repellency [38]. The processing techniques for creating superhydrophobic surfaces are diverse and include methods like electrospinning deposition, layer-by-layer assembly, chemical etching, phase separation, chemical vapor deposition (CVD), and colloid assembly [39,40,41,42,43,44].

One prevalent method for fabricating superhydrophobic surfaces is through direct laser texturing, which can create micro/nanostructures on metal surfaces without the need for chemical treatment [40,41,42]. This technique has been used to develop superhydrophobic aluminum alloy surfaces, where laser ablation parameters and laser spot sizes significantly influence the surface’s micro/nanostructures and, consequently, its wetting properties [41]. The fast wetting transformation from the hydrophilic to the superhydrophobic state on laser-textured stainless steel surfaces was investigated by Chun et al. [42], who used a crossed scanning laser beam to pattern the surface of SUS304 stainless steel, followed by undergoing a low-temperature annealing treatment at 100 °C (Figure 5).

Chemical etching is another cost-effective method that has been widely used to produce artificial superhydrophobic coatings. Subodh et al. [43] involve a two-step chemical etching process for fabricating the superhydrophobic/oleophobic microporous aluminum surface. Initially, the hierarchical microporous (HMP) structure was formed via chemical etching, followed by the application of a self-assembled monolayer of 1H,1H,2H,2H-perfluorooctyltrimethoxysilane (PFOTS) onto the aluminum substrate. This process resulted in a surface with superhydrophobic and oleophobic properties. The modified surface exhibited a water contact angle of 162°, confirming its superhydrophobic nature. It demonstrated oleophobic behavior against oils with surface tensions ranging from 21.5 to 63.3 mN/m. The surface retained its superhydrophobic characteristics after linear abrasion testing and exhibited self-cleaning, anti-fouling, and antibacterial properties.

The development of durable and stretchable superhydrophobic materials is also a significant area of research. For example, polymer nanofiber composites have been developed for efficient oil/water separation, incorporating hierarchical structures that mimic natural surfaces to achieve their unique properties [44]. Additionally, a novel superhydrophobic wood surface has been created using a PEG–functionalized SiO_2_/PVA/PAA/fluoropolymer hybrid transparent coating, which not only provides superhydrophobicity but also excellent durability and robustness (Figure 6) [45]. This approach utilizes vacuum physical evaporation and solution immersion techniques, requiring no prior wood pretreatment. Low surface energy is achieved solely through heating, without the need for any chemical modifiers, rendering the process both green and environmentally friendly.

In the realm of transparent superhydrophobic materials, research has shown their effectiveness in preventing dust accumulation on photovoltaic glass surfaces, thereby maintaining the power generation efficiency of photovoltaic modules [46]. The self-cleaning function of superhydrophobic materials has been identified as an important means to address the issue of ash accumulation in photovoltaic panels, and it has been proven feasible through various ash accumulation simulation experiments and field tests.

The processing of superhydrophobic materials involves a variety of techniques, each with its unique advantages and applications. Direct laser texturing and chemical etching are prominent methods that enable the creation of surfaces with specific micro- and nano-structures, endowing the materials with their characteristic water-repelling properties. These advanced materials have a broad range of potential applications, from improving the efficiency of oil/water separation to enhancing the durability and self-cleaning capabilities of various surfaces.

### 4.2. Methods of Paper-Based Materials Superhydrophobic Modification

#### 4.2.1. Surface Chemical Modification Technology

Surface chemical modification technology is a process that transforms the chemical composition and micro/nanostructure of paper surfaces to create superhydrophobic properties [23,39,47,48]. This technology enhances the water contact angle (WCA) to well above 150°, resulting in surfaces that are not only water-repellent but also resistant to other liquids, stains, and microbial adhesion [12,13,49]. By altering the paper’s surface chemistry through reactions that add functional groups or change the surface energy, this method allows for precise control over the surface properties at the molecular level [17,50,51,52,53]. This is crucial for applications that require specific wettability or reactivity, leading to improved self-cleaning properties, enhanced durability, and the potential for use in various applications such as food packaging, wearable electronics, and environmental protection [54]. The versatility and uniqueness of this technology are demonstrated by its ability to introduce new properties such as superhydrophobicity, antimicrobial activity, or enhanced sensing capabilities, making it a valuable tool in a wide range of industries [55,56,57,58,59,60].

Yu et al. [54] devised a method for creating durable, paper-based packaging material that exhibits superhydrophobic properties and possesses ethylene scavenging capability. (Figure 7). This material not only presented improved mechanical and barrier properties but also effectively extended the shelf-life of cherry tomatoes under ambient conditions, demonstrating its potential in sustainable food packaging. In addition, products related to this method demonstrate the potential of paper-based sensors in wearable devices for detecting human joint movements or gait, both in the air and underwater. He et al. [49] developed a simple spraying method to create superhydrophobic and conductive films on paper. The samples prepared with a 1:1 ratio of micro- to nano-graphite exhibited the best superhydrophobic properties with contact and rolling angles of 165.4° and 3.2°, respectively. These papers also maintained stable conductivity even underwater, opening up new possibilities for all-weather flexible electronic devices.

#### 4.2.2. Surface Coating Technology

Surface coating technology involves applying a layer onto the paper to impart new characteristics such as conductivity, superhydrophobicity, or barrier properties [14,61,62,63,64]. Surface coating technology that endows materials with superhydrophobic properties involves creating surfaces that repel water due to their micro- and nano-scale structures and low surface energy [15,65]. This method is versatile and can be used to create functional surfaces with tailored properties for specific applications. The coatings can range from conductive materials to superhydrophobic polymers, and they can significantly broaden the application scope of paper-based materials [66,67,68,69,70].

Xiang et al. [64] introduced a sustainable method for creating superhydrophobic cellulose nanocrystals (CNCs) by attaching octadecylamine (ODA) to their surface through the self-polymerization of tannic acid. The resultant ODA-PTA@CNCs can be applied via spray coating onto diverse substrates, including paper, cotton, and polyurethane, to form superhydrophobic layers. These coatings are not only sustainable and versatile but also demonstrate promising applications in waterproof paper-based packaging and anti-fouling textiles. The tensile strength of the filter paper was approximately 22.7 MPa in its dry state, but it significantly dropped to below 2.5 MPa when exposed to water. In contrast, the ODA-PTA@CNC-coated filter paper demonstrated water repellency, remaining unwetted even after being submerged in water for half an hour, and retained a tensile strength similar to that of the original dry filter paper.

Li et al. [15] developed a superhydrophobic and conductive paper using MXene nanosheets (Figure 8). This paper exhibits excellent EMI shielding capability, with a specific shielding effectiveness (SE) value of up to 3339.6 dB cm^2^ g^−1^. The material’s high conductivity also enables good Joule heating performance, making it suitable for wearable EMI shielding materials.

Another application is the creation of superhydrophobic and conductive films on paper. He et al. [63] fabricated micron/nano-graphite/polydimethylsiloxane (PDMS) coatings on filter paper surfaces using a straightforward spraying technique. These coatings demonstrate outstanding superhydrophobic and conductive characteristics, with optimal superhydrophobic performance achieved when the micro-graphite to nano-graphite ratio is 1:1. The samples exhibited contact and rolling angles of 165.4° and 3.2°, respectively.

#### 4.2.3. Physical Composite Technology

Physical composite technology combines different materials to create a composite with improved properties [71,72,73]. This method leverages the synergistic effects of material combinations to achieve performance that surpasses individual components. It allows for the development of paper-based materials with enhanced mechanical strength, electrical conductivity, or thermal management capabilities [74,75]. The physical composite technology for creating superhydrophobic surfaces on paper involves the integration of materials with distinct properties to achieve a surface that repels water. This technology is significant for various applications, including oil–water separation [76,77], self-cleaning surfaces [78,79], and anti-fouling coatings [80,81,82].

Xin Li et al. [76] developed a superhydrophobic paper using a breath figure method combined with nanoparticles. The synergistic effect of an amphiphilic copolymer of Poly(St-co-HEMA), nano silica particles (SiO_2_), and polydimethylsiloxane (PDMS) was utilized to create micro–nanostructures. The prepared superhydrophobic paper exhibited a static water contact angle of 165° and a sliding angle of 2°, demonstrating outstanding water repellency and self-cleaning ability. The material retained its superhydrophobic properties under various extreme conditions, such as exposure to low and high temperatures, strong acidic or alkaline environments, ultrasonic treatment, and repeated sandpaper abrasion. The paper demonstrated effective performance in separating water–oil mixtures, achieving an efficiency exceeding 90%, and remained reusable for at least 20 cycles.

Xue Liu et al. [75] developed a robust superhydrophobic paper by integrating silver nanowires and fluorinated titania nanoparticles using a conventional paper sizing agent (alkyl ketene dimer). This process formed a three-dimensional network structure on the paper surface. The resulting paper demonstrated outstanding water repellency, with a static contact angle of 165° and a roll-off angle of 6.2°. It also exhibited remarkable mechanical durability, retaining its superhydrophobic properties even after 130 cycles of linear sandpaper abrasion or high-velocity water jet impact. In addition to its superhydrophobic characteristics, the paper provided self-cleaning, electrical conductivity, and antibacterial functionalities, making it suitable for a wide range of applications.

Ning et al. [77] introduced a straightforward two-step immersion method to create superhydrophobic cellulose paper. The process involved direct modification of the paper with octadecyltrichlorosilane (OTS), followed by integration with hydrophobic silica. The resulting functionalized paper exhibited superhydrophobic properties, with a static water contact angle reaching up to 157°, and was also superoleophilic. It demonstrated resistance to acids, alkalis, and abrasion cycles. After undergoing 10 separation cycles for various heavy oil/water mixtures, the paper maintained a separation efficiency of 99.31% with fluxes exceeding 5100 L/(m^2^·h). This simple, scalable, and cost-effective superhydrophobic paper holds significant potential for applications in oil/water separation.

#### 4.2.4. Laser Etching Technology

Laser etching technology uses laser precision to create micro- and nanostructures on paper surfaces [83,84]. This method allows for the fabrication of paper-based analytical devices with high precision and control over channel dimensions and patterns. The patterns can be designed and modified on demand, enabling tailored functionalities for specific applications. And this method can be executed without relying on harsh chemicals, rendering it a more sustainable option [16]. Laser etching technology for hydrophobic surfaces on paper is a powerful method that combines precision, customization, and environmental sustainability.

Fan et al. [83] describe the preparation of a superwettable surface by laser etching, which combines polydimethylsiloxane (PDMS) and paper to create areas with superhydrophobic and superhydrophilic properties, as shown in Figure 9. The modified material features a surface with outstanding stability and flexibility, making it suitable for portable testing applications such as droplet collection, arrays, and colorimetric detection. He et al. [85] present a novel approach where TiO2-coated paper is modified to create superhydrophobic patterns through UV irradiation. The residual organics on μPADs can be easily bleached by TiO2-assisted degradation under UV or sunlight, indicating the dual functionality of the substrate for both sensing and self-cleaning.

#### 4.2.5. Other Methods

A variety of innovative techniques that do not fit into the above methods are generally based on the improvements of the previously mentioned centralized methods as well as their binary combinations [85,86,87]. These technologies showed innovative approaches to enhancing the hydrophobic properties of paper through biomimicry and nano/microstructure, offering eco-friendly, durable, and multifunctional solutions for a wide range of applications [88,89,90].

Liu et al. [85] developed a superhydrophobic paper-based strain sensor that functions not only as a wearable electronic device for monitoring human movements but also detects underwater vibrations. This innovation showcases its potential for applications in environmental protection and underwater robotics. This technology involves the fabrication of a paper-based strain sensor inspired by the micro/nanostructures of the lotus leaf and the scorpion’s slit sensillum. The sensor is created by fabricating microscale grooves on a paper substrate, sputtering Ag nanoparticles for enhanced conductivity, and applying a superhydrophobic coating made from a suspension of SiO_2_ nanoparticles and MWCNTs in a PDMS matrix. The sensor exhibits high sensitivity and water repellency with a contact angle of 164° and is capable of functioning in both wearable electronics and underwater applications. Similarly, Liu et al. [87] demonstrated a paper-based sensor that can monitor human joint or muscle motion and can be used for various tactile sensing applications under humid or aqueous conditions, showcasing its versatility and environmental adaptability. The sensor is fabricated by introducing V-shaped microgrooves inspired by arthropods into the paper substrate and coating it with a superhydrophobic layer inspired by plant surfaces. This paper-based material has a high strain sensitivity of 100, a pressure sensitivity of 0.43% kPa^−1^, and a strain resolution of 0.003%. It also exhibits superhydrophobicity with a water contact angle of up to 152.31° and a sliding angle of 7.31°.

#### 4.2.6. Comparison of Modification Methods

Each of the above methods has its unique approach to achieving superhydrophobicity, with distinct advantages and disadvantages. Table 1 provides a comprehensive comparison of various superhydrophobic modification methods for paper-based materials. This comparison aims to offer a clear overview of the different techniques, highlighting their key features and potential applications.

### 4.3. Application and Development of Superhydrophobic Paper-Based Materials

Superhydrophobic paper-based materials have been developed to leverage their unique properties in a variety of applications, ranging from conventional products to high-value functional materials. Superhydrophobic paper-based materials not only offer a range of functional advantages but also align with the growing global emphasis on sustainability. The eco-friendly nature of these materials is a key aspect that warrants further exploration [15,65]. The use of natural polymers such as cellulose and chitosan as base materials reduces the reliance on petroleum-based plastics. Additionally, many modification techniques, such as laser etching [83] and physical compositing [71,72,73,74,75], avoid the use of harsh chemicals, minimizing environmental pollution. The development of these materials using sustainable processes ensures that their production has a low environmental impact, making them a viable option for industries aiming to reduce their carbon footprint. One of the most significant advantages of paper-based materials is their biodegradability [13]. Unlike synthetic superhydrophobic materials that can persist in the environment for long periods, superhydrophobic papers can be designed to degrade naturally over time. This is particularly important in applications such as single-use diagnostics and packaging, where the material’s end-of-life disposal is a critical consideration. For example, paper-based sensors used in food testing can be disposed of without causing long-term environmental harm, as they will degrade naturally in the environment. The recyclability of superhydrophobic paper-based materials is another important factor contributing to their sustainability [64]. Paper is a widely recycled material, and the superhydrophobic modifications do not necessarily hinder its recyclability [65].

Self-Cleaning Material

Superhydrophobic paper-based materials exhibit self-cleaning properties, which are particularly useful in outdoor applications where surfaces are prone to dirt and grime accumulation [14,63,90,91,92]. The high water contact angle and low sliding angle allow water droplets to roll off the surface, carrying away dirt and contaminants (Figure 10) [54]. The PDMS/SA/HNTs@PDA@ paper demonstrated remarkable enhancements in mechanical and barrier properties. Concretely, the tensile strength of the modified paper increased by 89.9%, while the water vapor permeability decreased by 19.5% and the air permeability dropped by 27.5% in comparison to the original paper.

Oil–Water Separation

These materials have shown great potential in oil–water separation due to their superhydrophobic and superoleophilic nature [44,65]. They can selectively repel water while attracting oil, making them ideal for cleaning up oil spills and purifying industrial oily effluents. Research has indicated remarkable separation efficiency with oil purity values of ≥99.97 wt% and high permeation flux values observed in experiments. As illustrated in Figure 11, the study by Xiang et al. [65] shows that chloroform can pass through the ODA−PTA@CNC-coated cotton textile, while water stained with Rhodamine B is retained above the textile, indicating the effectiveness of the coating in separating oil from water. Toluene can be readily separated in a water-in-toluene emulsion and recovered by passing the emulsion through ODA−PTA@CNC-coated cotton textile, thus demonstrating the potential of ODA−PTA@CNCs in separating oil from water-in-oil emulsions.

Droplet Manipulation and Paper Fluidics

The unique wettability of these materials allows for precise control over droplet movement, which has applications in paper-based diagnostics and microfluidics [29,30]. They can be used to create heat-patterned paper fluidics for various analytical purposes. Wang et al. [61] employed a rigid–soft hybrid design approach to enhance the performance of paper-based pressure sensors. By incorporating bio-inspired microstructures into the sensor design, they achieved a device that exhibits rapid response and recovery times (less than 50 milliseconds), an extensive sensing range (up to 1 MPa), and remarkable cycling stability (over 5000 cycles).

Paper-Based Sensor

Paper-based sensors have emerged as a promising class of flexible and eco-friendly wearable devices, offering a wide range of applications in human health monitoring, environmental sensing, and smart robotics [78]. These sensors leverage the inherent properties of paper, such as its flexibility, biodegradability, and cost-effectiveness, to create functional devices that can detect various physical stimuli, including strain, pressure, and temperature changes [71,85].

One of the primary applications of paper-based sensors is in wearable electronics for monitoring human body movements and physiological signals. They can be integrated into electronic skin (E-skin) systems to detect spatial strain distribution on the skin during body movements, providing real-time feedback for health monitoring and sports performance analysis. For instance, paper-based strain sensors have been utilized to monitor finger bending, elbow joint motion, and even subtle facial expressions like smiling and drinking, demonstrating their potential to capture fine-scale muscular activities and gestures (Figure 12) [87].

In addition to strain sensing, paper-based sensors have also been developed for pressure detection, which allows them to function as tactile sensors and human–machine interfaces. They can accurately respond to touch and pressure variations, making them suitable for applications in prosthetics and robotics where sensitivity to touch is crucial [14]. Another significant application area is environmental monitoring, where paper-based sensors can be employed to detect changes in humidity, temperature, and even chemical substances. Their superhydrophobic properties and resistance to corrosion make them ideal for use in harsh outdoor conditions, including underwater applications. These sensors can maintain stable performance even when exposed to water, acids, alkalis, and various organic solvents, expanding their potential use in environmental sensing and industrial processes [13].

These applications highlight the versatility and potential impact of superhydrophobic paper-based materials in various industries. As research continues to advance the fabrication and modification techniques, it is expected that these materials will find even more applications in the future.

## 5. Summary

The realm of superhydrophobic paper-based functional materials is gaining significant attention for their unique properties and wide applicability. These materials, inspired by natural superhydrophobic surfaces like the lotus leaf, are created by combining low-surface-energy materials with micro/nanostructures that promote air pocket formation, resulting in high water contact angles and low sliding angles. The paper-based materials, known for their renewability, biodegradability, and sustainability, are modified to achieve superhydrophobicity, thereby expanding their utility in applications such as oil–water separation, anti-corrosion, and self-cleaning.

The development of superhydrophobic paper-based materials is poised to expand further due to ongoing research and innovation in fabrication and modification techniques. As these materials become more refined, their applications are expected to diversify, potentially infiltrating new markets such as aerospace for de-icing surfaces, construction for self-cleaning building materials, and textiles for waterproof clothing. The integration of sensor technology within these materials could also lead to smart textiles that monitor health parameters or environmental sensors that provide real-time data on pollution levels.

Furthermore, the push towards sustainability and green technologies will likely accelerate the adoption of these materials, as they offer eco-friendly solutions compared to traditional synthetic materials. The potential for upscaling production using papermaking technology could make these materials more cost-effective and accessible, driving their integration into mainstream applications. The future of superhydrophobic paper-based materials looks promising, with the potential to revolutionize various industries and contribute significantly to environmental conservation and technological advancement.

## Figures and Tables

**Figure 1 nanomaterials-15-00107-f001:**
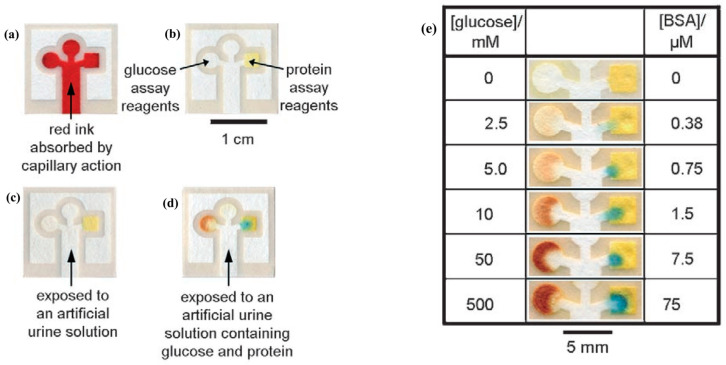
Chromatography paper patterned with photoresist developed by Martinez et al. Reproduced with permission [6]. Copyright 2007, John Wiley and Sons. (**a**) Patterned paper after absorbing Waterman red ink (5 μL) by capillary action. The central channel absorbs the sample by capillary action and the pattern directs the sample into three separate test areas. (**b**) Complete assay after spotting the reagents. The square region on the right is the protein test and the circular region on the left is the glucose test. The circular region on the top was used as a control well. (**c**) Negative control for glucose (left) and protein (right) by using an artificial urine solution (5 μL). (**d**) Positive assay for glucose (left) and protein (right) by using a solution that contained 550 mM glucose and 75 μM BSA in an artificial urine solution (5 μL). The control well was spotted with the potassium iodide solution, but not with the enzyme solution. (**e**) Glucose and protein detection assays by using varying concentrations of glucose and BSA.

**Figure 2 nanomaterials-15-00107-f002:**
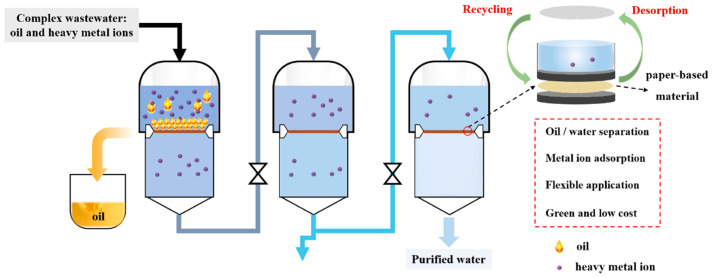
Illustration of the adsorption and separation process utilizing a multifunctional paper-based material. Reproduced with permission [31]. Copyright 2022, Elsevier.

**Figure 3 nanomaterials-15-00107-f003:**
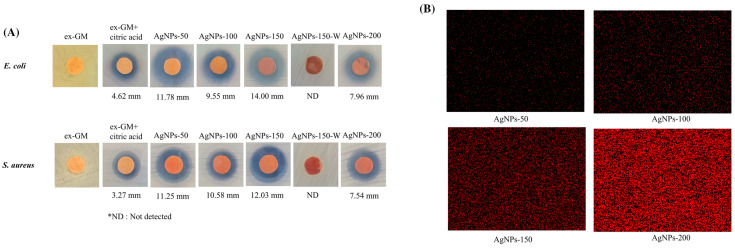
Smart packaging with antibacterial effects. (**A**) Antibacterial activities of paper coated with AgNPs and (**B**) EDS images of coated paper with AgNPs [36].

**Figure 4 nanomaterials-15-00107-f004:**
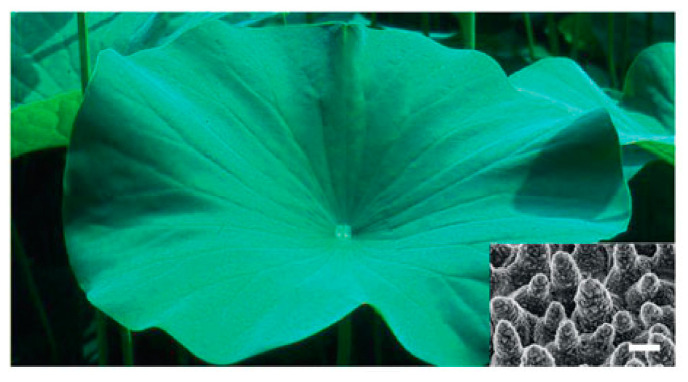
The lotus leaf, renowned for its remarkable water-repellent properties facilitated by its hierarchical micro/nanostructures (see inset image). Scale bar = 10 μm [38].

**Figure 5 nanomaterials-15-00107-f005:**
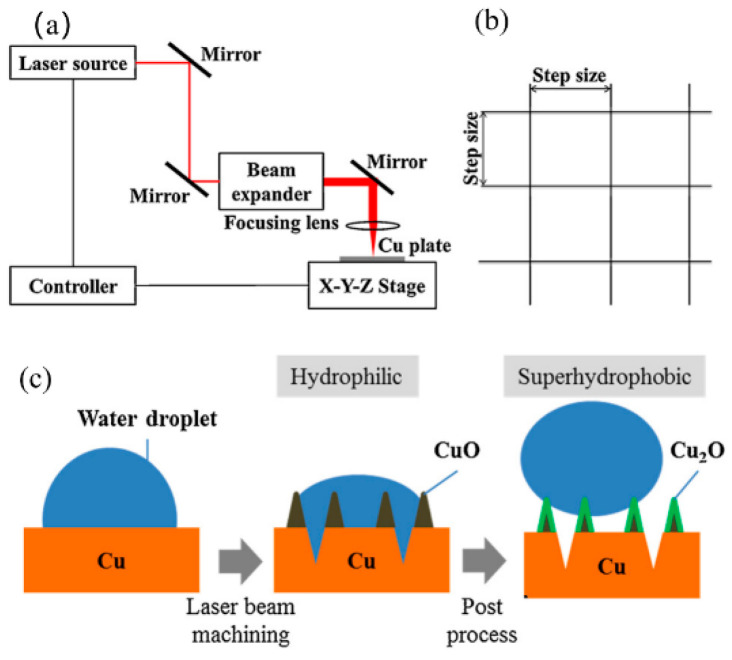
Illustrative diagrams of (**a**) the laser beam machining system, (**b**) the beam path design, and (**c**) the mechanism for achieving superhydrophilic and superhydrophobic surfaces through laser beam machining followed by post-processing. Reproduced with permission [42]. Copyright 2016, Elsevier.

**Figure 6 nanomaterials-15-00107-f006:**
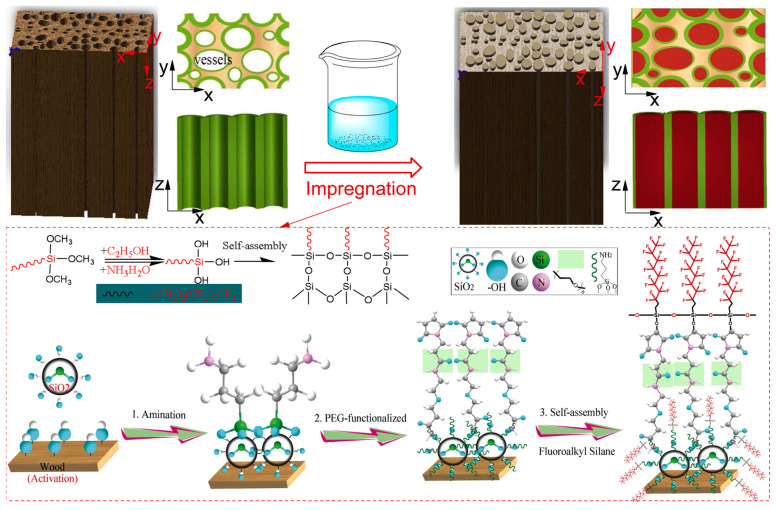
Illustration of processing approach of robust superhydrophobic wood. Reproduced with permission [45]. Copyright 2021, Elsevier.

**Figure 7 nanomaterials-15-00107-f007:**
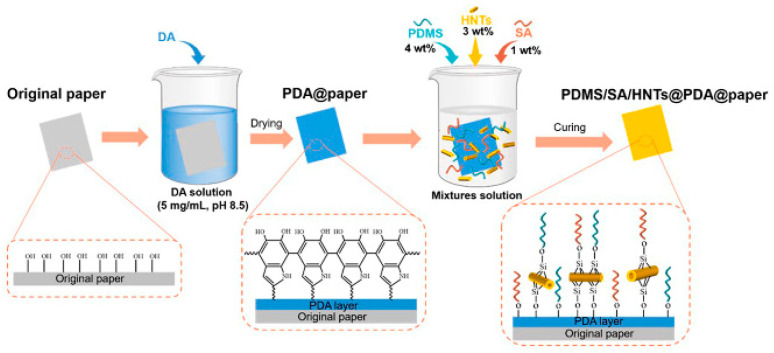
Illustrative diagram depicting the preparation process of PDMS/SA/HNTs@PDA@paper composite material. Reproduced with permission [54]. Copyright 2023, Elsevier.

**Figure 8 nanomaterials-15-00107-f008:**
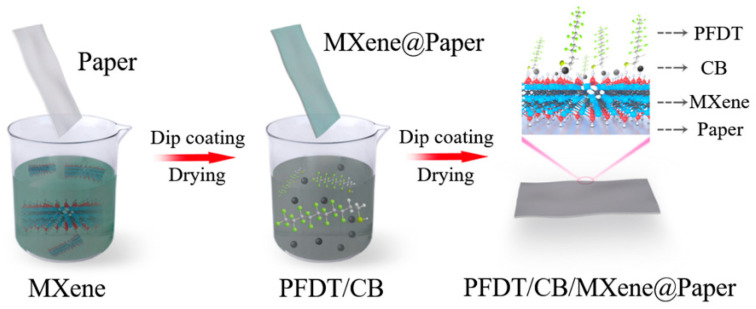
Schematic illustration of the preparation process of PFDT/CB/MXene@Paper. Reproduced with permission [15]. Copyright 2022, Royal Society of Chemistry.

**Figure 9 nanomaterials-15-00107-f009:**
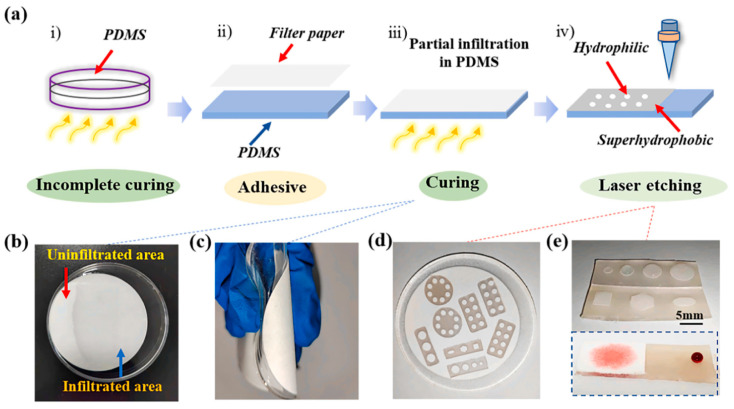
The flexible patterned surfaces with special wettability obtained by simple laser etching. Reproduced with permission [83]. Copyright 2022, Elsevier. (**a**) Schematic illustration of the preparation process for the bionic super-wetting surface. (**b**) Filter paper demonstrating PDMS infiltration and non-infiltration at the bottom. (**c**) The filter paper adheres firmly to the PDMS once it is fully cured. (**d**,**e**) Surfaces with various geometric shapes created using laser selective etching.

**Figure 10 nanomaterials-15-00107-f010:**
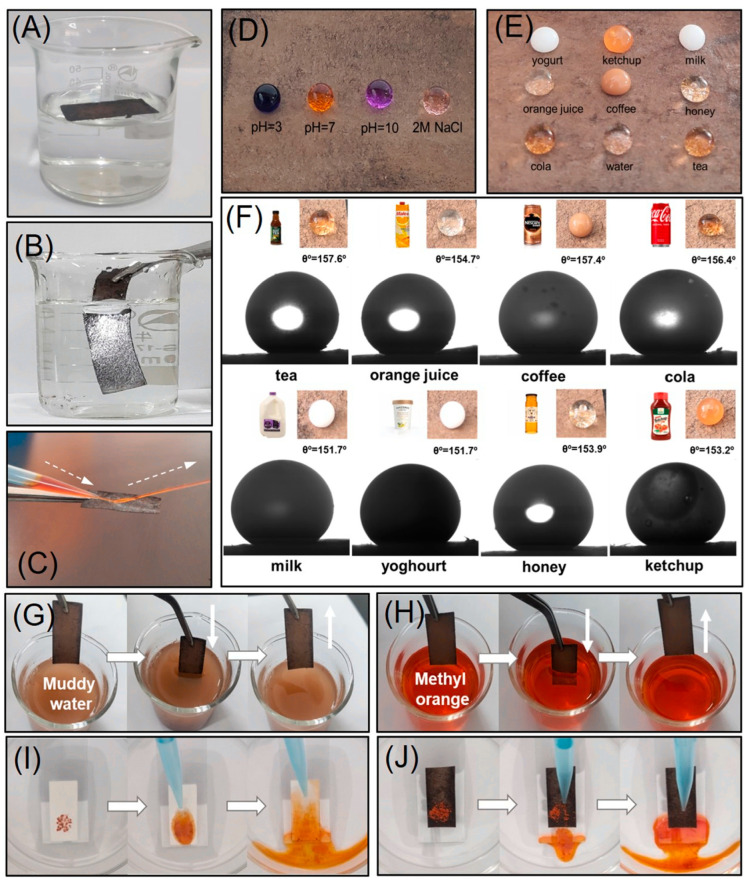
Self-cleaning properties of hydrophobic paper-based materials. Reproduced with permission [54]. Copyright 2023, Elsevier. (**A**) An optical image showing the superhydrophobic paper floating on the water’s surface. (**B**) A silver mirror effect observed when the paper surface is submerged in water due to external force. (**C**) The rebound of a water jet from the paper surface. (**D**,**E**) Optical images depicting various liquid droplets on the prepared paper surface. (**F**) Images illustrating the water contact angles (WCA) of different liquid food droplets on the prepared paper surface. (**G**,**H**) The anti-fouling test procedures for superhydrophobic paper in both muddy water and methyl orange-dyed water. (**I**,**J**) Comparison of the self-cleaning performance between untreated and superhydrophobic paper surfaces through testing processes.

**Figure 11 nanomaterials-15-00107-f011:**
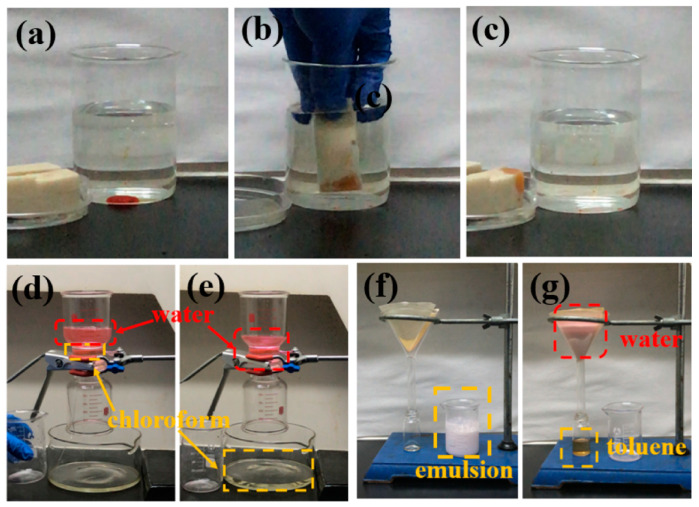
The ODA−PTA@CNC-coated sponge could absorb oil (chloroform) at the bottom of water and remove it from the aqueous medium. Reproduced with permission [65]. Copyright 2022, American Chemical Society. (**a**–**c**) Oil-absorbing behavior of ODA-PTA@ CNC-coated PU sponges. (**d**,**e**) ODA-PTA@ CNC-coated cotton cloth was employed for oil–water separation via filtration. (**f**,**g**) ODA-PTA@ CNC-coated paper was used for oil–water separation from water-in-oil emulsion vial filtration.

**Figure 12 nanomaterials-15-00107-f012:**
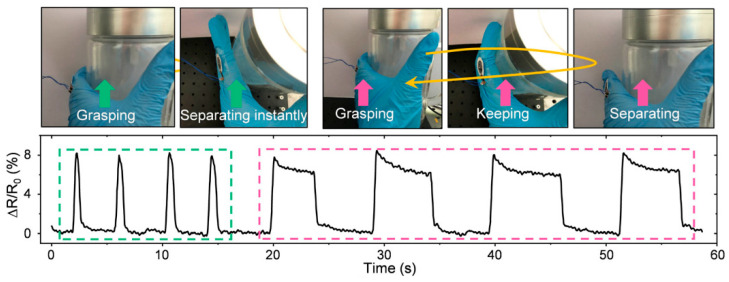
The response curve of the paper-based sensor serving as a strain sensor for bottle-grasping motion, including grasping then separating instantly and grasping for a while before separating. Reproduced with permission [87]. Copyright 2013, Royal Society of Chemistry.

**Table 1 nanomaterials-15-00107-t001:** Comparison of superhydrophobic modification methods for paper-based materials.

Method	Description	Advantages/Disadvantages	References
Surface Chemical Modification Technology	This process transforms the chemical composition and micro/nanostructure of paper surfaces to create superhydrophobic properties.	Allows precise control over surface properties at the molecular level, crucial for specific wettability or reactivity. The chemical modification process can be complex, requiring specific chemicals and reaction conditions. Higher safety requirements for the environment and operators.	[12,13,17,23,39,47,48,49,50,51,52,53,54,55,56,57,58,59]
Surface Coating Technology	Involves applying a layer onto the paper to impart new characteristics.	Versatile and can be used to create functional surfaces with tailored properties for specific applications. Significantly broadens the application scope of paper-based materials. The adhesion and durability of the coating may be an issue. The selection and preparation of the coating material are critical and can affect the final performance.	[14,15,62,63,64,65,66,67,68,69,70]
Physical Composite Technology	Combines different materials to create a composite with improved properties.	Leverages the synergistic effects of material combinations to achieve performance that surpasses individual components. Enables the integration of superhydrophobic surfaces with other functionalities. The fabrication process can be complex, requiring precise control of the proportion and mixing of different components. The compatibility and interfacial bonding between different materials can affect the overall performance of the material.	[71,72,73,74,75,76,77,78,79,80,81]
Laser Etching Technology	Uses laser precision to create micro- and nanostructures on paper surfaces.	Allows for high-precision, customized manufacturing of paper-based analytical devices with specific channel dimensions and patterns. A more sustainable option as it does not rely on harsh chemicals. Requires professional laser equipment and operation skills. The etching process may cause some damage to the internal structure of the paper.	[16,83,84]
Other Methods	Various innovative techniques that are generally based on improvements of the previously mentioned methods or their binary combinations.	Provides more innovative ideas and methods for achieving multifunctional integration. Offers the potential to develop more environmentally adaptable and practical superhydrophobic paper-based materials. These methods are relatively new and may still be in the research and exploration phase, with immature processes. The universality and reproducibility of some methods need further verification.	[85,86,87,88,89,90]

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
