# Peer review of "Advancements in Superhydrophobic Paper-Based Materials: A Comprehensive Review of Modification Methods and Applications"

_nanomaterials, 2025, doi:10.3390/nano15020107_

Round 1
Reviewer 1 Report
Comments and Suggestions for Authors
Advancements in Superhydrophobic Paper-Based Materials: A Comprehensive Review of Modification Methods and Applications
This paper presents the review of Advancements in Superhydrophobic Paper-Based Materials. While the manuscript summarizes relevant data and represent comprehensive work, several areas could be clarified to enhance quality and readability.
1.. Other properties of superhydrophobic paper-based materials should be presented (mechanical, wettability, tear length, absorption test according to Klem, etc.). Relation between superhydrophobicity versus other parameters.
2. Eco-friendliness, biodegradability and recyclability.
3. In general, this manuscript suffers from additional figures and schemes. Advantages and drawbacks of mostly important/potential processes should be highlighted.
4. Minor English editing is necessary.
Author Response
Comments 1.. Other properties of superhydrophobic paper-based materials should be presented (mechanical, wettability, tear length, absorption test according to Klem, etc.). Relation between superhydrophobicity versus other parameters.
Response 1 Thank you for pointing this out. We agree with this comment. Your advice is very important and we have ignored the contrast between hydrophobicity and other physical properties. Such as the PDMS/SA/HNTs@PDA@ paper demonstrated remarkable enhancements in me-chanical and barrier properties. Concretely, the tensile strength of the modified paper increased by 89.9%, while the water vapor permeability decreased by 19.5% and the air permeability dropped by 27.5% in comparison to the original paper. The ODA-PTA@CNC-coated filter paper demonstrated water repellency, remaining un-wetted even after being submerged in water for half an hour, and retained a tensile strength similar to that of the original dry filter paper.
This part has been adjusted in the revised manuscript.
Comments 2. Eco-friendliness, biodegradability and recyclability.
Response 2 Thank you for pointing this out. We agree with this comment. Superhydrophobic paper-based materials not only offer a range of functional advantages but also align with the growing global emphasis on sustainability. The eco-friendly nature of these materials is a key aspect that warrants further exploration[64,65]. The use of natural polymers such as cellulose and chitosan as base mate-rials reduces the reliance on petroleum-based plastics. Additionally, many modification techniques, such as laser etching[83,83] and physical compositing[71~75], avoid the use of harsh chemicals, minimizing environmental pollution. The development of these materials using sustainable processes ensures that their production has a low environmental impact, making them a viable option for industries aiming to reduce their carbon footprint. One of the most significant advantages of paper-based materials is their biodegradability[46]. Unlike synthetic superhydrophobic materials that can persist in the environment for long periods, superhydrophobic papers can be designed to degrade naturally over time. This is particularly important in applications such as single-use diagnostics and packaging, where the material's end-of-life disposal is a critical consideration. For example, paper-based sensors used in food testing can be disposed of without causing long-term environmental harm, as they will degrade naturally in the environment. The recyclability of superhydrophobic paper-based ma-terials is another important factor contributing to their sustainability[62]. Paper is a widely recycled material, and the superhydrophobic modifications do not necessarily hinder its recyclability[64].
Comments 3. In general, this manuscript suffers from additional figures and schemes. Advantages and drawbacks of mostly important/potential processes should be highlighted.
Response 3 Your comments are very constructive. We agree with this comment. The corresponding table has been added to the revised manuscript.
Comments 4. Minor English editing is necessary.
Response 4 Thank you for your valuable feedback. We agree with this comment. We have done our best to adjust the English for receiving manuscripts.
Reviewer 2 Report
Comments and Suggestions for Authors
The review is very well written and it is easy to read. It represents also a valuable contribution to the field. Thus, my opinion is that should be published as it is.
Author Response
Thank you very much for your valuable comments
Reviewer 3 Report
Comments and Suggestions for Authors
I think that the topic of this review paper is interesting and that it is well written .
Author Response

(The authors gave the same response as above.)

Reviewer 4 Report
Comments and Suggestions for Authors
Before publishing the manuscript, it is necessary to make the following corrections:
1. In the introduction, it is necessary to emphasize the importance of the research, highlighting what is new.
2. Highlight in the introduction the difference with the articles already published, for example:
a) https://doi.org/10.1016/j.ccr.2021.214207
b) 24 https://doi.org/10.1002/advs.202308152
3. Add a title to the table that appears in figure 1.
4. Expand the information in the food testing section
Author Response
Comments 1. In the introduction, it is necessary to emphasize the importance of the research, highlighting what is new.
Response 1: Thank you very much for your insightful comments on our manuscript. We have carefully considered your suggestions and highlighted the novel aspects of our work, clearly delineating how it contributes to the existing body of knowledge and why it is important in the context of the field. We believe these changes will provide a stronger foundation for our study and better engage the readers.
Comments 2. Highlight in the introduction the difference with the articles already published, for example:
- a) https://doi.org/10.1016/j.ccr.2021.214207
- b) 24 https://doi.org/10.1002/advs.202308152
Response 2: Thank you for your valuable feedback. In accordance with your suggestion, we have revised the introduction to highlight the differences between our study and the previously published articles, including https://doi.org/10.1016/j.ccr.2021.214207 and https://doi.org/10.1002/advs.202308152.
We have carefully analyzed the content of these articles and compared them with our research. And the relevant articles are listed in the references to highlight the article. In the introduction, we have clearly outlined the unique aspects of our study, emphasizing how our approach, methodology, and findings differ from those presented in the aforementioned articles. With the development of new technologies and new materials, the structure and processing methods of hydrophobic materials are also constantly updated. For example, the application of materials with special conductivity and surface chemical properties such as MXene in superhydrophobic paper based materials makes the application of this material. We believe that these revisions will provide a clearer context for our research and highlight its originality and significance.
Comments 3. Add a title to the table that appears in figure 1.
Response 3:Thank you for pointing this out. We agree with this comment. The corresponding diagram description has been added to the revised manuscript.
Comments 4. Expand the information in the food testing section
Response 4 Thank you for pointing this out. We agree with this comment. The application of paper-based sensors in the realm of food testing is an bur-geoning domain, offering cost-effective and user-friendly solutions for the detection of food contaminants[27]. Furthermore, the development of paper-based sensors is not confined to detecting a single type of pollutant.
In practical applications, the portability and ease of use of paper-based sensors make them ideal for on-site rapid testing. They can be designed as disposable test strips or portable detection kits, allowing food producers, regulators, and consumers to conduct quick tests anywhere. However, despite the great potential of paper-based sensors in food testing, there are still challenges to overcome. Enhancing the sensitivity and specificity of sensors to ensure accurate detection of low-concentration contami-nants is a crucial research direction. Additionally, developing sensors that can operate stably in complex food matrices, as well as improving production efficiency and re-ducing costs, are also key focuses of current research. With continuous technological advancements, paper-based sensors are expected to play an increasingly important role in food safety and quality control. These sensors can be customized to identify specific pollutants such as pesticide residues and microbial contamination. For instance, re-searchers have developed paper-based electrochemical biosensors for detecting ethanol levels in beer[28], as well as inkjet-printed flexible biosensors for the rapid, label-free detection of antibiotics in milk[29]. These applications underscore the practical value of paper-based sensors in ensuring food safety and quality control.